# MGCFNN: A Neural MultiGrid Solver with Novel Fourier Neural Network for High WaveNumber Helmholtz Equations

**Yan Xie,**[*] **Minrui Lv& Chen-song Zhang**
State Key Laboratory of Mathematical Sciences,
Academy of Mathematics and Systems Science,
Chinese Academy of Sciences, Beijing 100190, China;
School of Mathematical Sciences,
University of Chinese Academy of Sciences, Beijing 100049, China.
`{xieyan2021, lvminrui, zhangcs}@lsec.cc.ac.cn`

## Abstract

Solving high wavenumber Helmholtz equations is notoriously challenging. Traditional solvers have yet to yield satisfactory results, and most neural network methods struggle to accurately solve cases with extremely high wavenumbers within heterogeneous media. This paper presents an advanced multigrid-hierarchical AI solver, tailored specifically for high wavenumber Helmholtz equations. We adapt the MGCNN architecture to align with the problem setting and incorporate a novel Fourier neural network (FNN) to match the characteristics of Helmholtz equations. FNN, mathematically akin to the convolutional neural network (CNN), enables faster propagation of source influence during the solve phase, making it particularly suitable for handling large size, high wavenumber problems. We conduct supervised learning tests against numerous neural operator learning methods to demonstrate the superior learning capabilities of our solvers. Additionally, we perform scalability tests using an unsupervised strategy to highlight our solvers' significant speedup over the most recent specialized AI solver and AI-enhanced traditional solver for high wavenumber Helmholtz equations. We also carry out an ablation study to underscore the effectiveness of the multigrid hierarchy and the benefits of introducing FNN. Notably, our solvers exhibit optimal convergence of $\mathcal{O}(k)$ up to $k \approx 2000$.

## 1 Introduction

### 1.1 Background

The Helmholtz equation, a fundamental partial differential equation (PDE), is pivotal in describing wave propagation phenomena across various disciplines, including acoustics, electromagnetics, and seismology. Despite its importance, solving the Helmholtz equation presents significant challenges due to the high wavenumber and variable wave speed. These factors necessitate a large number of grid points to accurately capture the wave shape and result in a complex-valued, highly ill-conditioned, indefinite system. Indeed, it is widely acknowledged that the optimal number of iterations required for a solver with complexity $\mathcal{O}(N)$ to solve a high wavenumber $k$ Helmholtz equation is $\mathcal{O}(k)$. This is due to the complex propagation pattern of the wave in heterogeneous media, with a slow decay rate.

Numerous research efforts have been dedicated to developing efficient traditional solvers, such as the Born series method Osnabrugge et al. (2016), the factorization method Osei-Kuffuor & Saad (2010), the shifted Laplacian method Gander et al. (2015); Sheikh et al. (2013); Calandra et al. (2013), domain decomposition method Chen & Xiang (2013); Leng & Ju (2022; 2019), and multigrid method Brandt

---

[*]Corresponding author.

& Livshits (1997). However, these methods have not yet achieved satisfactory results Ernst & Gander (2011).

Data-driven methods have recently emerged as a popular approach to solving PDEs, either by optimizing certain solver components or by learning the entire solver. One significant research area is Physics Informed Neural Networks (PINNs) Raissi et al. (2019), which use neural networks to approximate PDEs' solution functions. While recent works Song et al. (2022); Escapil-Inchauspé & Ruz (2023) have applied PINNs to Helmholtz equations, they do not address high wavenumber cases. Furthermore, our focus is on inference rather than online training to gain the solution.

A more related research area is neural operator learning, which learns the operator (mapping) between the infinite-dimensional parameter and solution spaces of PDEs. This includes works such as the Convolutional Neural Operators (CNO) Raissi et al. (2019), Deep Operator Network (DeepONet) Lu et al. (2021), Fourier Neural Operator (FNO) Li et al. (2021), U-NO Rahman et al. (2023), and MgNO He et al. (2024). These methods solve PDEs during the inference phase, functioning more like a solver for discretized PDEs. The MgNO He et al. (2024) work has tested their method on Helmholtz equations and compared it with other neural operator learning methods, but did not consider high wavenumber cases.

## 1.2 Related Works

In practice, it is nearly impossible to directly solve high wavenumber Helmholtz equations within heterogeneous wave speed (or slowness) field to a desired accuracy with just a single application of the network (see Subsection 4.2). Actually, both traditional and AI solvers require more iterations when dealing with higher wavenumber. Recent studies Azulay & Treister (2022); Cui et al. (2022); Stachenfeld et al. (2021); Stanziola et al. (2021) have developed iterative AI solvers to tackle this challenging problem, capable of achieving higher accuracy through iteration. However, the wavenumbers tested in these studies remain limited. Our objective is to solve high wavenumber Helmholtz equations up to $k \approx 2000$, aiming to match the application scope of traditional solvers, but with a significantly improved speed.

Several latest studies have made significant progress in this area. The study Lerer et al. (2024) introduced a pure AI solver, termed encoder-solver, and the work Cui et al. (2024) presented a dedicated AI-enhanced traditional multigrid solver framework, Wave-ADR-NS, for Helmholtz equations. Both of them demonstrated superior performance over traditional shifted Laplacian solvers. However, the encoder-solver relies on an existing traditional solver for data generation and solve phase assistance. Meanwhile, Wave-ADR-NS requires extensive problem insights to construct a solver with good convergence, limiting its applicability to other wave equations.

Moreover, the MGCNN work Xie et al. (2023) laid out principles for building an efficient AI solver for discretized linear PDEs, guaranteeing quick convergence and adaptability to different iterative frameworks for desired accuracy. Despite these advantages, MGCNN has not been applied to solve the Helmholtz equation, as its architecture is not directly suitable for this specific problem.

## 1.3 Our Contributions

In this work, we propose a multigrid-hierarchical AI solver that effectively addresses the challenges of solving high wavenumber Helmholtz equations. The innovations and contributions are listed as follows:

- We significantly modify the spectral convolution operation in the FNO to create the Fourier Neural Network (FNN), which handles *full-mode* information, making it suitable for high wavenumber problems and scalable for larger problem sizes.
- The FNN, which functions mathematically equivalent to a CNN with an extended kernel size, facilitates wave propagation. We adapt the MGCNN to address high wavenumber Helmholtz equations and incorporate the FNN into its solve phase. This integration forms the MGCFNN solver, which combines the complementary features of the novel FNN and the multigrid hierarchy, resulting in improved convergence and shorter solve time.
- In contrast to recent specialized iterative AI solvers for high wavenumber Helmholtz equations, our solvers do not necessitate specialized domain knowledge. This makes them easy

to train and utilize, with the potential to directly tackle a broader range of wave equations. Notably, the proposed MGCFNN achieves at least $4.8\times$ speedup compared with other AI iterative solvers under the same problem settings.

- Although our solvers are designed for unsupervised learning, we also conduct supervised experiments. Our solver demonstrates superior learnability compared to other operator learning networks, achieving lower error with shorter training time and fewer parameters.

## 2 PRELIMINARIES

### 2.1 FORMULATION AND DISCRETIZATION

The Helmholtz equation is expressed as

$$- \Delta u(x) - \left( \frac{\omega}{c(x)} \right)^2 (1 - \gamma i)u(x) = f(x), \;\; x \in [0,1]^2, \tag{1}$$

where $u$ represents the unknown field, $c$ is the wave speed, $\omega$ is the angular frequency, and $f$ is the source term. In line with related works Azulay & Treister (2022); Lerer et al. (2024); Cui et al. (2024), we define $\kappa(x) = \frac{1}{c(x)}$ as the wave slowness and refer to $k = \frac{\omega}{c} = \omega\kappa$ as the wavenumber. $i = \sqrt{-1}$ is the imaginary unit, and $\gamma$ is the damping factor, indicating the medium's wave absorption property. This work primarily focuses on the hardest case of $\gamma = 0$, while also comparing with work Lerer et al. (2024) under $\gamma = 0.01$ in our experiments. As for boundary conditions, we employ absorbing boundary conditions Engquist & Majda (1979); Erlangga et al. (2006) to reduce wave reflection at the boundary, simulating radiation in an infinite domain. Specifically, we set $\gamma$ to increase quadratically from 0 to 1 at a certain distance from the boundary.

Applying a second-order finite difference discretization to the Helmholtz equation 1 on a uniform grid with spacing $h$ yields the linear system

$$A_h u_h = f_h, \tag{2}$$

where $A_h$ represents the discretized Helmholtz operator, expressed in stencil form as

$$A_h = \frac{1}{h^2} \begin{bmatrix} 0 & -1 & 0 \\ -1 & 4 - \omega^2\kappa^2(x)h^2 & -1 \\ 0 & -1 & 0 \end{bmatrix} \tag{3}$$

and $f_h$ is the discretized source term. To accurately capture the wave shape, the spacing $h$ must be sufficiently small, typically requiring at least 10 grid points per wavelength. [1] This implies that $\omega\kappa h \leq \frac{2\pi}{10}$. Consequently, we increase the grid size $1/h$ in our experiments to accommodate the increased wavenumber $k$. If multiplying both sides of the equation by the factor $\frac{1}{h^2}$ and denoting $\omega\kappa h$ along with $\gamma$ as the problem coefficient $coef$, we find that $coef$ is scale-invariant, serving as a normalized input to the network for different scales.

### 2.2 FOURIER TRANSFORM

The Fourier transform is a mathematical operation that decomposes a function into harmonic wave components with different frequencies, making it particularly well-suited for wave propagation problems. In the frequency domain, a globally propagated wave with a specific frequency can be simply represented within a circle. From another perspective of view, the convolution theorem Smith (1997) states that the convolution of two functions in the spatial domain corresponds to the pointwise multiplication of their Fourier transforms in the frequency domain, i.e.,

$$\mathcal{F}(f * g) = \mathcal{F}(f)\mathcal{F}(g), \tag{4}$$

where $\mathcal{F}$ denotes the Fourier transform. Consequently, the Fourier transform can be utilized to perform a convolution operation with an extended kernel size, enabling the rapid propagation of wave source influence during the solve phase. As we use Fast Fourier Transform in our experiments, we denote FFT as the Fourier transform operation and IFFT as the inverse Fourier transform operation.

---

[1] To illustrate high wavenumber cases and keep the comparison experiments feasible, we may break the rule to use high wavenumber with relatively few grid points.

## 3 NETWORK ARCHITECTURE

The architecture comprises a setup phase network for managing problem coefficients and a solve phase network that utilizes the handled coefficient information to process the source term, also known as the right-hand-side term (see Appendix C), to propose a solution. We use multigrid hierarchy to build the two phases, and design a novel FNN kernel to handle the high frequency wave propagation in the coarse levels of the solve phase.

### 3.1 FOURIER NEURAL NETWORK

While the Fourier Neural Operator (FNO) Li et al. (2021) demonstrates impressive learnability across a range of PDEs, it encounters limitations when solving high wavenumber Helmholtz equations. We make significant modifications to FNO's spectral convolution operation.

- We separate the processing of the coefficient and source term, ensuring the network linearly operates the source terms, as will be shown in the level status process equation 5.

- We assume that the weights form a smooth function in the frequency domain (see Fig. 1a), and thus use a smaller weight tensor to interpolate weights across the entire frequency domain. This approach ensures high-frequency modes are captured within a limited parameter size. This step is denoted as *case 1* in Fig. 2.

- To apply the FNN to problems with larger size and higher wavenumber, we first perform IFFT to obtain the convolutional kernel in the space domain, then pad it to the size of the larger target problem domain, and finally apply FFT to obtain the weights in the frequency domain. This step is denoted as *case 2* in Fig. 2.

- Moreover, the *input* tensor is interpreted as the concatenation of real and imaginary parts of a complex tensor, halving the number of channels prior to the operation. Conversely, the *output* tensor channels are doubled from a complex tensor. These decrease the computational cost of the Fourier transform while maintaining the network's predictive performance.

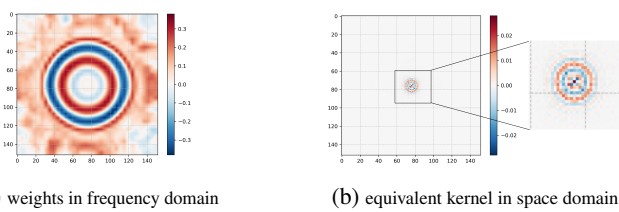

(a) weights in frequency domain      (b) equivalent kernel in space domain

Figure 1: The setuped weight tensor of FNN.

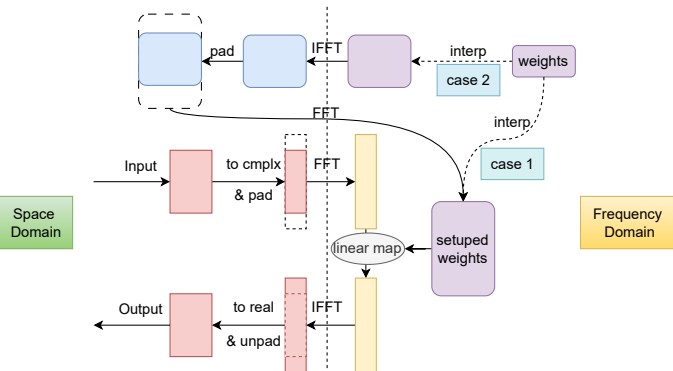

Figure 2: The network architecture of FNN. *Case 1*: Interpolated weights are of the same size as the target problem. *Case 2*: The target problem is larger than the interpolated weights. Box width is proportional to the number of channels, and box height is proportional to the number of grid points.

**Remark 3.1.** *For higher wavenumber problems, we initially planned to leverage the invariance of the function in the frequency domain (see Appendix A) and apply the interpolation technique again to obtain the weight tensor. However, a straightforward interpolation method performs poorly for substantially larger wavenumber. The mathematical equivalence in equation 4 aids us in proposing a more elegant and efficient approach.*

**Remark 3.2.** *FNN is mathematically connected to the inverse of the damped Helmholtz operator, as detailed in Appendix A. Additionally, the role it plays in solve phase is mathematically related to the traditional Born series method Osnabrugge et al. (2016), which solves the equation with an iterative pseudo-propagation process. A detailed discussion on this connection is provided in Appendix B.*

In summary, the network architecture of the FNN is depicted in Fig. 2, and its detailed pseudocode is provided in Appendix D. It's important to note that for a given problem size, FNN first performs interpolation or even additional IFFT and FFT to set up the weight tensor. This is a one-time cost for all same-sized problems.

**Effective Kernel Size.** Despite FNN's utilization of all Fourier transform modes, its effective influence length is limited, as illustrated in Fig. 1b. On one hand, fixed-length padding, which corresponds to the effective kernel size of the equivalent convolution operation, is adequate. On the other hand, a multigrid hierarchy is still crucial to handle problems on multiple levels to accelerate influence transmission when the multiple grids can resolve the high frequency wave, as shown in Subsection 4.4. We then introduce the multigrid-hierarchical setup and solve phase network to efficiently address the challenge in conjunction with FNN.

## 3.2 SETUP PHASE

The setup phase processes the coefficients of the PDEs, including the wave speed $c(x)$ (or slowness $\kappa(x)$) and the damping factor $\gamma(x)$, preparing necessary information for each level in solve phase. We streamline the setup phase of MGCNN from Xie et al. (2023) by integrating the downsampling and nonlinear ResCNN processing into a single workflow (see Alg. 1). The setup phase only needs to be executed once for fixed PDE coefficients and multiple right-hand-side tensors, as is the case for the updated residual terms in an iterative solving framework (see Appendix C).

---

**Algorithm 1** Setup Phase Network

1: INPUT: $coef$ is problem coefficient tensor, $level$ is number of levels for multigrid hierarchy.
2: OUTPUT: $setup\_outs$ is a list of setup tensors for each level.
3: **procedure** SETUP($coef, level$)
4:     $setup\_out_1 \leftarrow reChannelCNN(coef)$
5:     **for** $l$ in $1, 2, \ldots, level$ **do**
6:         $setup\_out_l \leftarrow NonLinearResCNN(setup\_out_l)$       ▷ Nonlinear processing
7:         **if** $l < level$ **then**
8:             $setup\_out_{l+1} \leftarrow RestrictCNN(setup\_out_l)$       ▷ Downsampling
9:         **end if**
10:     **end for**
11:     **return** $setup\_outs \leftarrow [setup\_out_1, setup\_out_2, \ldots, setup\_out_{level}]$
12: **end procedure**

---

## 3.3 SOLVE PHASE

The solve phase framework of MGCFNN closely follows that of MGCNN as described in Xie et al. (2023). The key distinction lies in the replacement of the CNN components in ResNets at certain coarse levels with our custom-designed FNN. During both the down and up cycles, each level involves three primary steps:

1. Utilize ResNet (either ResCNN or ResFNN) in conjunction with the setup tensor at the current level to process the status tensor. Specifically, we have

$$x_l = ResNet(setup\_out_l, x_l) = Net(setup\_out_l \cdot x_l) + x_l. \tag{5}$$

2. During the down cycle, after processing, employ the restriction operator to transfer the status tensor to the next coarser level.

3. During the up cycle, before processing, add the prolonged status tensor from the coarser level.

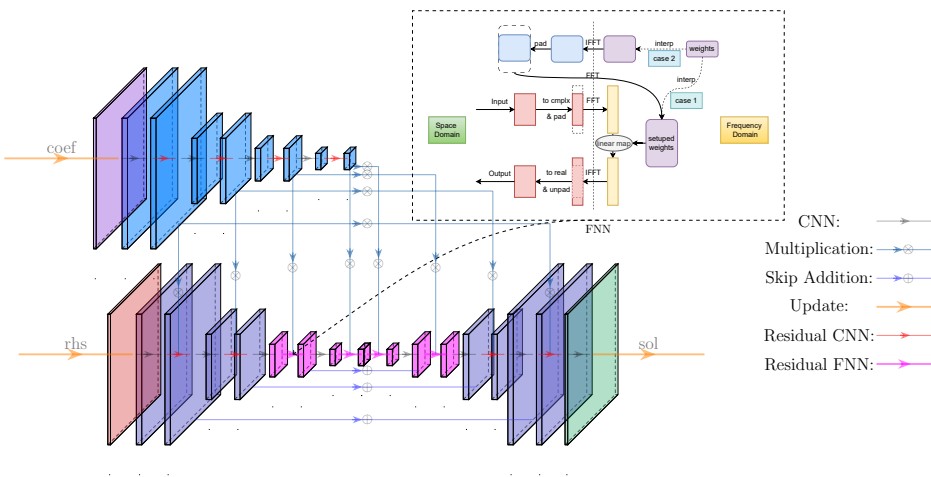

Figure 3: The network architecture of MGCFNN. Its coarsest two levels use FNN kernels.

By combining the aforementioned setup and solve phase networks, we arrive at our novel solver, MGCFNN, as shown in Fig. 3.

**Architecture Naming.** We use the following naming conventions for our multigrid-hierarchical networks: MGCFNN refers to networks that utilize FNN in some coarse levels of the solve phase. MGFNN refers to networks that use FNN at all levels. MGCNN refers to networks that use CNN at all levels.

**Multigrid Challenges.** As demonstrated in previous literature Xie et al. (2023), MGCNN can function as a global operator, using more levels with weight sharing across levels to handle problems of larger sizes. However, applying this strategy to higher wavenumber cases in our scalability test settings presents two challenges. First, the grid must be sufficiently large to accommodate the high wavenumber. Actually, it is common practice to maintain fixed few number of levels of the multigrid hierarchy in traditional methods Calandra et al. (2013) for higher wavenumber. Second, the coarse grid problem faces the same wavenumber with fewer grid points, resulting in dissimilar properties across different levels.

**Architecture Techniques.** In response to these challenges, besides our efforts of developing FNN to transmit wave influence more globally, we use fixed number of levels of the multigrid hierarchy in our experiments and increase the network's computational complexity when moving to coarser level. Specifically, number of levels are determined by number of points per wavelength and we double the number of sweeps (layers) for all of our multigrid-hierarchical solvers on coarser level. The detailed architecture settings are provided in Appendix H.

**Remark 3.3.** *The settings of network are closely related to frequency. MGCFNN replaces the CNN with FNN at levels where the grid size is nearly insufficient to resolve high-frequency waves but still too large for CNN to handle. From one perspective, using FNN at all levels is unnecessary, as shown in Subsection 4.4. From another perspective, there is no need to use FNN for low-frequency waves, where more coarse grids can still resolve waves. Therefore, the optimal use of FNN is problem-dependent, which is a common feature for iterative solvers.*

In summary, we combine the complementary features of the novel FNN and the specialized multigrid hierarchy to develop the MGCFNN solver, which effectively handles high wavenumber Helmholtz equations.

## 4 NUMERICAL EXPERIMENTS

### 4.1 BASIC SETTINGS

**Comparison Methods.** We first compare our solvers with well-known operator learning methods to demonstrate our networks' learnability. While there are some Graph Neural Network (GNN) based methods Trifonov et al. (2024); Li et al. (2023), they are designed for unstructured problems, do not utilize structural features, and are typically computationally slower than CNNs. Operator learning methods are not directly suitable for iterative applications to improve accuracy, so we only perform a single inference to compare results in Subsection 4.2. Some strategies Azulay & Treister (2022); Lerer et al. (2024); Rudikov et al. (2024); Kopaničáková & Karniadakis (2025) exist to improve the accuracy of these methods, such as using flexible Krylov methods for nonlinear preconditioning and generating suitable right-hand-side (source) terms for training. In Subsection 4.3, we compare our work with AI, AI-enhanced traditional solvers and sparse direct solver.

**Loss function & Data.** Both supervised and unsupervised training strategies are applicable. However, we recommend unsupervised learning as it eliminates the need for labeled data generation. Supervised learning is only employed in Subsection 4.2 to facilitate a fair comparison with other operator learning methods under identical settings. The loss functions are the averaged square of the $L_2$ norm of error and residual terms of the linear equations respectively:

$$\begin{cases} L_{err} = \frac{1}{N} \sum_{i=1}^{N} ||u_i - sol_i||_2^2, & \text{supervised learning}, \\ L_{res} = \frac{1}{N} \sum_{i=1}^{N} ||rhs_i - A_i sol_i||_2^2, & \text{unsupervised learning}, \end{cases} \tag{6}$$

where $N$ is the data count, $u_i$ is the true solution, $rhs_i$ is the input source term, $sol_i$ is the output guessed solution, and $A_i$ is the discretized linear operator. The operator is determined by the coefficient tensor $coef_i$, which includes the normalized slowness field $\kappa$ and $\gamma$. We generate $rhs_i$ as a white noise tensor to cover all frequency ranges. The $\kappa$ field is generated using the CIFAR-10 Krizhevsky et al. (2009) dataset, with images interpolated to the target problem size and linearly transformed to range $[\kappa_{\min}, \kappa_{\max}]$. In practice, $coef_i$ should be generated to fit specific application scenarios. We also present results from an ultrasonic CT scenario in Subsection 4.4, which are more representative of real-world applications. Furthermore, Appendix G shows generalization results on STL-10 Coates et al. (2011) dataset and the benchmark Marmousi Brougois et al. (1990) data, both of which are of higher resolution.

Although the solve phase network can be applied several times when solving in an iterative framework, we only perform the solve phase once for each $coef_i$ during training.

**Default Settings.** Unless stated otherwise, experiments follow these defaults: slowness bounds $\kappa_{\min}$ and $\kappa_{\max}$ are 0.25 and 1.0 respectively. Solve test results represent the median of 50 unseen data. Training is unsupervised, executed on a RTX 4090 GPU using PyTorch with CUDA 12.4. Both training and inference use float32 precision, while the solver employs a mixed precision strategy C to achieve rtol=1E-7 tolerance for the norm of relative residual. We refer to the solver within a stationary iterative framework as a standalone solver and also show the results of using GMRES to speed up the convergence. For scalability tests, we only train our models on a $511 \times 511$ grid and test on grid size up to $4095 \times 4095$.

Note that larger models are used in small size problems to fully utilize GPU resources, while smaller models are employed in scalability tests to fit within GPU memory constraints. For detailed hyperparameters, refer to Appendix H.

### 4.2 COMPARISON OF DIFFERENT SOLVERS

To provide a fair illustration of our solvers' learnability, we employ supervised learning, a training strategy commonly used in other operator learning methods. As most operator learning methods are not designed for use in an iterative framework, we compare the results after a single network inference. We limit these tests to a grid size of $256 \times 256$ and use a relatively high angular frequency $\omega = 80\pi$ to set high wavenumber $k$. We compare our methods with recent well-known operator learning methods, including MgNO He et al. (2024), FNO2d Li et al. (2021), U-NO Rahman et al. (2023), Dil-ResNet Stachenfeld et al. (2021), LSM Wu et al. (2023), MWT2d Gupta et al. (2021), and

U-Net Ronneberger et al. (2015). We utilize the official implementations of these solvers, adjusting the tunable parameters to suit the problem and balance the train error, train time and parameter size. All solvers are trained under identical settings (refer to Appendix I). To assess the model's ability to generalize, we record the point *Where the Best Test Error* is achieved. This includes the test error, the corresponding training error, and the epoch index within the total of 120 epochs.

We decrease the value of $\kappa_{\min}$ to aggravate the heterogeneity of the media, and separately evaluate models for $\kappa_{\min} = 0.75, 0.5, 0.25$. As demonstrated in Table 1 and Table 2, our proposed models, MGCFNN and MGFNN, outperform other solvers in terms of train and test error, time efficiency, and parameter size, which represent the state-of-the-art in this field. Notably, while models like FNO2d, MWT2d, and U-NO can achieve certain low training errors, they are prone to high test errors and early overfitting. This further underscores the superior generalization and robustness of our models.

Table 1: Supervised learning on grid $256 \times 256$ with $\omega = 80\pi$ and $\kappa_{\min} = 0.75$. We record information *Where the Best Test Error* reaches.

| Model | Train Error | $\kappa_{\min} = 0.75$ | | | Train Time(s/epoch) | Parameters(MB) |
|---|---|---|---|---|---|---|
| | | Where Best Test Error | | | | |
| | | Test Error | Train Error | Epoch | | |
| MGFNN | **0.035** | **0.061** | **0.035** | 120 | 93.2 | 8.9 |
| MGCFNN | 0.046 | 0.070 | 0.046 | 120 | **67.5** | 5.3 |
| MGNO | 0.063 | 0.079 | 0.063 | 120 | 110.3 | 4.6 |
| FNO2D | 0.085 | 0.561 | 0.496 | 4 | 103.6 | 46.1 |
| MWT2D | 0.119 | 0.527 | 0.475 | 4 | 147.0 | 26.0 |
| U-NO | 0.408 | 0.880 | 0.870 | 4 | 101.6 | 86.7 |
| U-NET | 0.534 | 0.803 | 0.758 | 31 | 89.9 | 31.0 |
| DIL-RESNET | 0.605 | 0.606 | 0.605 | 116 | 140.0 | **0.6** |
| LSM | 0.722 | 0.783 | 0.739 | 66 | 230.7 | 4.9 |

Table 2: Supervised learning on grid $256 \times 256$ with $\omega = 80\pi$ and $\kappa_{\min} = 0.50, 0.25$. We record information *Where the Best Test Error* reaches.

| Model | $\kappa_{\min} = 0.50$ | | | | $\kappa_{\min} = 0.25$ | | | |
|---|---|---|---|---|---|---|---|---|
| | Train Error | Where Best Test Error | | | Train Error | Where Best Test Error | | |
| | | Test Error | Train Error | Epoch | | Test Error | Train Error | Epoch |
| MGFNN | **0.075** | **0.187** | 0.122 | 25 | **0.126** | **0.431** | **0.253** | 18 |
| MGCFNN | 0.101 | 0.206 | **0.104** | 88 | 0.190 | 0.458 | 0.396 | 7 |
| MGNO | 0.209 | 0.333 | 0.221 | 81 | 0.361 | 0.633 | 0.614 | 14 |
| FNO2D | 0.122 | 0.749 | 0.718 | 3 | 0.154 | 0.851 | 0.818 | 3 |
| MWT2D | 0.169 | 0.728 | 0.669 | 4 | 0.186 | 0.845 | 0.821 | 3 |
| U-NO | 0.389 | 0.871 | 0.864 | 3 | 0.362 | 0.935 | 0.930 | 4 |
| U-NET | 0.634 | 0.774 | 0.716 | 57 | 0.575 | 0.806 | 0.778 | 26 |
| DIL-RESNET | 0.627 | 0.629 | 0.627 | 117 | 0.699 | 0.702 | 0.699 | 119 |
| LSM | 0.852 | 0.860 | 0.852 | 117 | 0.867 | 0.878 | 0.874 | 38 |

These results, especially in Table 2, suggest that applying a neural network just once to achieve the desired accuracy for high wavenumber and heterogeneous Helmholtz equations may not be feasible. In subsequent experiments, we will concentrate solely on iterative solving. Moreover, in scalability tests, solvers are expected to handle higher angular frequency up to $\omega = 640\pi \approx 2000$.

## 4.3 SCALABILITY COMPARISON

In this paper, we place greater emphasis on the scalability to larger problems with higher wavenumber. The solver should be capable of iteratively achieving any desired level of accuracy. The latest noteworthy iterative AI solvers to handle the challenging high wavenumber Helmholtz equations include pure AI solver Lerer et al. (2024), termed as encoder-solvers, and the Wave-ADR-NS solver, an AI-enhanced traditional multigrid solver Cui et al. (2024). Both methods have shown superior convergence and solve time performance compared to the traditional shifted Laplacian solver. However, we are unable to replicate the encoder-solver results, and the Wave-ADR-NS, while effective, is complex to implement and not currently open source. Therefore, we use their published results for comparison under the same data and hardware settings. Furthermore, while direct solvers are potentially less scalable to 3D problems than iterative methods, they are a viable option for challenging 2D problems. We compare our results with the sparse LU solver CHOLMOD Davis & Hager (2009); Chen et al. (2008); Davis et al. (2004); Amestoy et al. (1996; 2004) in the SuiteSparse Davis (2024) software. Its GPU acceleration Rennich et al. (2014) is not beneficial in our problems, so we run CHOLMOD

with single precision on CPU (Intel(R) Xeon(R) Gold 6458Q). The results are presented in Table 3, and our proposed MGCFNN significantly outperforms other solvers.

Table 3: Scalability comparison with other solvers.

| rtol=1E-7 | | $\gamma = 0.01$ (RTX3090) Lerer et al. (2024) | | | $\gamma = 0.0$ (A100 80G) Cui et al. (2024) | | | $\gamma = 0.0$ (CPU) | |
| --- | --- | --- | --- | --- | --- | --- | --- | --- | --- |
| | | time (s) & iters | | | time (s) & iters | | | time(s) | |
| $\omega$ | grid | MGCFNN | ENCODER-SOLVER | speedup | MGCFNN | WAVE-ADR-NS | speedup | CHOLMOD | speedup |
| $80\pi$ | $511 \times 511$ | 0.12(9) | 0.65(43) | 5.5 | 0.16(14) | 15.07(28) | 94.8 | 0.49 | 3.1 |
| $160\pi$ | $1023 \times 1023$ | 0.19(11) | 1.29(68) | 6.8 | 0.30(22) | 34.98(54) | 116.2 | 8.88 | 29.5 |
| $320\pi$ | $2047 \times 2047$ | 0.58(14) | 3.40(85) | 5.8 | 1.15(40) | 91.63(122) | 79.6 | 38.91 | 33.8 |
| $640\pi$ | $4095 \times 4095$ | 2.77(18) | 13.34(117) | 4.8 | 8.55(83) | 286.14(247) | 33.5 | 183.61 | 21.5 |

## 4.4 ABLATION STUDY

Firstly, we examined the FNN, MGFNN, and MGCFNN models on two datasets with fixed grid sizes. The first dataset has a grid size of $256 \times 256$ and an angular frequency of $\omega = 80\pi$. The second dataset, from an ultrasonic CT competition[2], has a grid size of $480 \times 480$ and an angular frequency of $\omega = 150\pi$. Table 4 presents the results for random sources, while Fig. 4b and Fig. 4a show single source results for illustration. As shown in Table 4, the multigrid hierarchy allows MGFNN to converge faster than FNN, but its use of FNN on all levels only reduces a few iterations compared to MGCFNN. MGCFNN solves problems faster in terms of time and is expected to be comparably faster for larger problems. Moreover, MGFNN uses more memory than MGCFNN, making it less suitable for large-scale problems.

Table 4: Model architecture ablation study.

| rtol=1E-7 | grid size $256 \times 256$, $\omega = 80\pi$ | | | | | | grid size $480 \times 480$, $\omega = 150\pi$ | | | | | |
| --- | --- | --- | --- | --- | --- | --- | --- | --- | --- | --- | --- | --- |
| | standalone | | | GMRES | | | standalone | | | GMRES | | |
| | MGCFNN | MGFNN | FNN | MGCFNN | MGFNN | FNN | MGCFNN | MGFNN | FNN | MGCFNN | MGFNN | FNN |
| iters | 22 | 19 | 55 | 15 | 13 | 24 | 12 | 9 | 54 | 11 | 10 | 22 |
| time(s) | 0.136 | 0.135 | 0.270 | **0.111** | 0.116 | 0.151 | **0.093** | 0.101 | 0.724 | 0.105 | 0.124 | 0.365 |

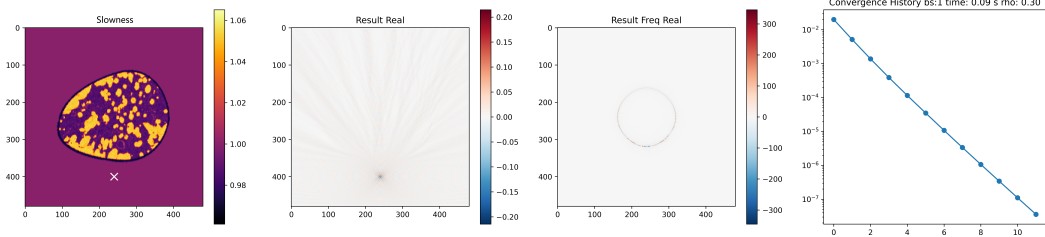

(a) Single source wave with $\omega = 80\pi$, in a slowness field of a cell.

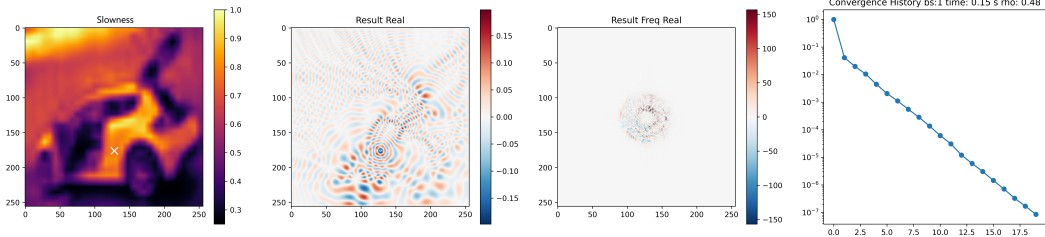

(b) Single source wave with $\omega = 150\pi$, in a slowness field generated from a CIFAR-10 data.

Figure 4: From left to right: slowness field $\kappa(x)$, real value of wave field and wave field in frequency domain, iterative solving history. The cross symbol in the first column denotes the position of source.

Secondly, we assess the scalability of MGCFNN and MGCNN across varying grid sizes and angular frequencies. Detailed results are in Table 5. Both models show optimal $\mathcal{O}(k)$ convergence for high

---

[2]AI4S Cup - Prediction of Wavefield in Ultrasonic Computed Tomography

wavenumber $k$, whether as standalone solvers or preconditioners in the GMRES solver. Notably, MGCFNN performs effectively as a standalone solver. In contrast, MGCNN requires 2 to 3 times more iterations and relies on GMRES to significantly reduce its iteration count. This confirms MGCFNN's effective learning of the Helmholtz equation's inverse operator. Despite the unexpected computation inefficiency of linear map in frequency domain (see Appendix E), MGCFNN still outperforms MGCNN in solve time.

Table 5: Scalability comparison between MGCFNN and MGCNN.

| rtol=1E-7 | | standalone | | | | GMRES | | | |
|---|---|---|---|---|---|---|---|---|---|
| | | MGCFNN | | MGCNN | | MGCFNN | | MGCNN | |
| $\omega$ | size | iters | time(s) | iters | time(s) | iters | time(s) | iters | time(s) |
| $80\pi$ | $511 \times 511$ | 14 | 0.17 | 35 | 0.27 | 12 | 0.17 | 21 | 0.20 |
| $160\pi$ | $1023 \times 1023$ | 22 | 0.31 | 61 | 0.55 | 20 | 0.31 | 36 | 0.37 |
| $320\pi$ | $2047 \times 2047$ | 41 | 1.18 | 115 | 2.33 | 35 | 1.13 | 71 | 1.65 |
| $640\pi$ | $4095 \times 4095$ | 83 | 8.18 | 231 | 18.66 | 72 | 8.48 | 146 | 14.56 |

**Remark 4.1.** *The solve time for the largest problem is approximately 8.2 seconds. This represents a significant improvement over recent parallel traditional solvers, such as source transfer Leng & Ju (2019) and trace transfer Leng & Ju (2022) methods, which typically require over fifty seconds to handle similar problems.*

## 5 CONCLUSIONS

We modify MGCNN architecture and further introduce a novel network component, the Fourier Neural Network (FNN), to improve performance for high wavenumber Helmholtz equations. FNN manages the information of full modes and scales to larger size and higher wavenumber problems. In standard supervised learning scenarios, MGCFNN surpasses other operator learning methods, showcasing superior learnability. Focusing on iterative solving for any desired accuracy and scalability to larger, higher wavenumber problems, MGCFNN significantly outperforms the most recent specialized AI solver and AI-enhanced traditional solver for Helmholtz equations and sparse direct solver CHOLMOD. The ablation studies further demonstrate the benefits of multigrid hierarchy and the hybrid model pattern, verifying the complementary features of FNN and multigrid hierarchy. Both MGCFNN and modified MGCNN exhibit an optimal convergence property of $\mathcal{O}(k)$ for high wavenumber $k$ up to approximately 2000, and MGCFNN excels in terms of number of iterations and solve time. All the experiments establish MGCFNN as the state-of-the-art AI solver for high wavenumber Helmholtz equations. Moreover, it reaches a high wavenumber scope comparable to the advanced parallel traditional solvers, while significantly reducing solve time.

In the present study, the interpolation operation in FNN remains simplistic, and the properties and best practices of FNN for solving Helmholtz equations are not yet fully explored. Furthermore, we have not addressed various boundary conditions, including reflection boundary conditions, which may conflict with the periodic boundary effect of the Fourier transform or the padding technique. These limitations highlight areas for future exploration and improvement.

**Reproducibility Statement.** The code and configuration scripts used in our experiments are available at `https://gitee.com/xiehuohuo77/mgcfnn/`. Our models can be trained using unsupervised learning, and we recommend this approach. To reproduce our results, you need some open datasets for speed (or slowness) fields. The speed data from the Ultrasonic CT area is sourced from the AI4S Cup, and we provide the Marmousi data tested in Appendix G. Other datasets are directly accessible through the provided PyTorch code.

**Acknowledgements.** This research is supported by the Strategic Priority Research Program of the Chinese Academy of Sciences, Grant No. XDB0640000, XDB0640300.

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

## A  FNN WEIGHTS IN FREQUENCY AND SPACE DOMAINS

Consider the weights of the FNN in the frequency domain and their equivalent kernels in space domain at each level of a MGFNN, as depicted in Fig. 5 and Fig. 6. The weights in the frequency domain appear smooth, and in the space domain, they exhibit a larger kernel size than common CNNs. These observations align with our design assumptions. Moreover, the weights in the space domain behave like a damped wave propagation.

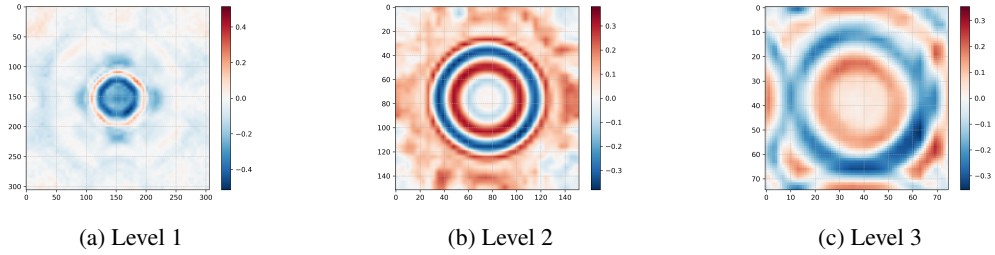

|           (a) Level 1           |           (b) Level 2           |           (c) Level 3           |

Figure 5: The weights of FNN in frequency domain.

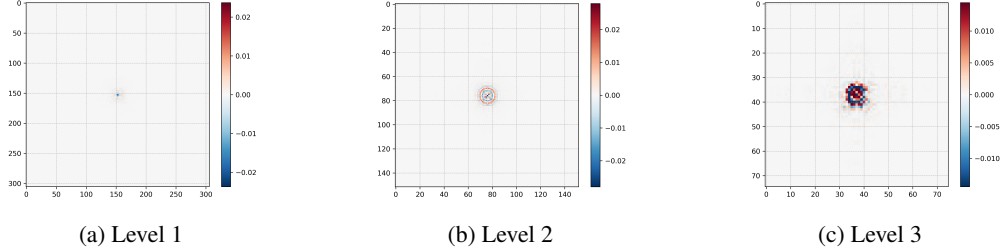

(a) Level 1  (b) Level 2  (c) Level 3

Figure 6: The equivalent kernels of the weights of FNN in space domain.

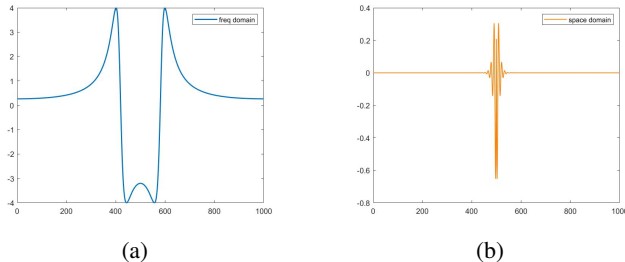

(a)  (b)

Figure 7: The inverse of the damped Helmholtz operator in frequency domain and its corresponding weights in space domain. $\gamma = 0.5, \omega\kappa h = 0.5$ and $N = 1000$.

To further elucidate this, we examine the expression of the inverse of a damped Helmholtz operator in the frequency domain. The one-dimensional stencil of the damped Helmholtz operator can be expressed as follows:

$$[-1, 2 - k_{h\gamma}^2, -1], \tag{7}$$

where $k_{h\gamma}^2 = \omega^2\kappa^2h^2(1 - \gamma i)$. For a mode $e^{\frac{-2mn\pi}{N}i}$ with frequency $m$, where $n$ is the grid index and $N$ is the grid size, the inverse of the damped Helmholtz operator on this mode is:

$$\frac{1}{2(1 - cos(\frac{2m}{N}\pi)) - k_{h\gamma}^2} e^{\frac{-2mn\pi}{N}i}. \tag{8}$$

If $k_{h\gamma}^2$ remains invariant with the increase of $N$, and $m$ increases at the same rate, this operation result also remains *invariant*. Given a large damping factor $\gamma = 0.5$, we can view the inverse of the damped Helmholtz operator in the frequency domain in Fig. 7a and its corresponding weights in the space domain in Fig. 7b. Similar patterns are observed in both the frequency and space domains. Thus, we can interpret FNN as learning a damped Helmholtz operator in the frequency domain.

**Remark A.1.** *We observe that the weights in the frequency domain are not smooth enough, which is expected since a wave with a certain frequency will exhibit centralized energy in the frequency domain. As shown in Fig. 5, the circles appear suboptimal with some serration resulting from a simple interpolation operation. This is a potential area for improvement in future work.*

## B  CONNECTION WITH BORN SERIES METHOD

The Born series method Osnabrugge et al. (2016) uses the Green's function theorem to solve the Helmholtz equation in a heterogeneous medium. Specifically, for a Helmholtz equation

$$\nabla^2 u(x) + k(x)^2 u(x) = -f(x), \quad x \in \mathbb{R}^d, \tag{9}$$

we can rewrite it in a damped form as

$$\nabla^2 u(x) + (k_0^2 - i\epsilon)u(x) = -f(x) - (k(x)^2 - k_0^2 + i\epsilon)u(x). \tag{10}$$

Suppose $g_0$ is the Green's function of the damped Helmholtz operator, and denote $G = \mathcal{F}^{-1} g_0 \mathcal{F}$. Further, denote $V$ as the diagonal matrix of $k(x)^2 - k_0^2 + i\epsilon$. We can rewrite Eq. 10 as

$$u = GVu + Gf. \tag{11}$$

The Born series method solves Eq. 11 iteratively as a series expansion of the operator $1/(1 - GV)$, i.e.,

$$u = \sum_{n=0}^{\infty} (GV)^n Gf = (1 + GV + (GV)^2 + \cdots)Gf. \tag{12}$$

This method can be iteratively written as

$$u^{k+1} = GVu^k + Gf, \tag{13}$$

which is similar to our solve phase in FNN as

$$x_l = FNN(setup\_out_l \cdot x_l) + x_l. \tag{14}$$

Recall that FNN performs like a damped Helmholtz operator, $setup\_out_l$ contains information from $k(x)$ and $x_l$ is the current state, starting from $f$ initially. Thus, the FNN in our solve phase level can be interpreted as a learnable Born series method for solving the Helmholtz equation.

## C  ITERATIVE FRAMEWORK AND MIXED PRECISION STRATEGY

In the context of a linear system $A\,sol = rhs$, where $A$ is a square matrix, $sol$ is the solution vector, and $rhs$ is the right-hand-side vector, the objective of a linear solver is to find the solution $sol$ that meets satisfactory tolerance which can be obtained iteratively enhancing the solution's accuracy.

One such iterative method is the *stationary iterative method*, which solves linear systems by iteratively addressing the error equation $A\,sol = r$, where $r = rhs - A\,sol$ represents the residual vector. This method can be defined as follows:

$$sol^{k+1} = sol^k + B\,r^k = sol^k + B\,(rhs - A\,sol^k), \tag{15}$$

where $sol^k$ is the solution at the $k$-th iteration and $B$ serves as the solver, providing an approximate error correction for each residual $r^k$. In our experiments, our neural network functions as the solver $B$ in the iterative framework.

The Krylov subspace method Golub & Van Loan (2013) is another iterative framework where the solver $B$ acts as a preconditioner. This method often speeds up the iteration process, albeit with some overheads. In this category, we use the Generalized Minimal Residual (GMRES) method and apply a restart technique to prevent storage of excessively large subspace vectors and to reduce the computations required for subspace orthogonalization. In our experiments, we set the restart number to 10, unless convergence is achieved rapidly, in which case it is set to 5.

To attain high accuracy, we employ a straightforward *mixed precision* strategy for both iterative frameworks. The computationally intensive part, either the stationary iteration's solver network $B$ or the entire GMRES solve process before restart, operates in float32 precision. To avoid error accumulation, the residual is updated in float64 precision.

## D  FNN PSEUDOCODE

FNN operates in two phases: a setup phase (Alg. 2) to initialize the weight tensor in the frequency domain, and a solve phase (Alg. 3) to perform spectral convolution. Note that this setup phase should not be confused with the setup phase network described in Subsection 3.2, which takes the problem coefficients as input.

---

**Algorithm 2** FNN setup phase

---

1: INPUT: $size$ is the size of the input tensor of the FNN.
2: OUTPUT: $setup\_weight$ is the tensor of the FNN weights in frequency domain to do linear mapping.
3: **procedure** SETUPFNNWEIGHT($size$)
4:     $init\_weight \leftarrow$ loaded or initialized
5:     $interp\_size \leftarrow$ loaded or settled
6:     **if** $size = interp\_size$ **then**                              ▷ case 1
7:         $setup\_weight \leftarrow interpolate(init\_weight)$
8:     **else**                                                          ▷ case 2
9:         $space\_weight \leftarrow \mathcal{F}^{-1}(init\_weight)$
10:        $space\_weight \leftarrow$ pad $space\_weight$ to $size$
11:        $setup\_weight \leftarrow \mathcal{F}(space\_weight)$
12:    **end if**
13:    **return** $setup\_weight$
14: **end procedure**

---

---

**Algorithm 3** FNN spectral convolution

---

1: INPUT: $input$ is the input tensor of the FNN, $setup\_weight$ is the tensor of the FNN weights in frequency domain.
2: OUTPUT: $output$ is the output tensor of the FNN.
3: **procedure** FNNCONV($input, setup\_weight$)
4:     $C \leftarrow$ number of channels of the input tensor
5:     $input\_cmplx \leftarrow input[: C/2] + input[C/2 :] \cdot i$          ▷ real to complex value
6:     pad $input\_cmplx$
7:     $input\_freq \leftarrow \mathcal{F}(input\_cmplx)$
8:     $output\_freq \leftarrow linear\_map(input\_freq, setup\_weight)$
9:     $output\_cmplx \leftarrow \mathcal{F}^{-1}(output\_freq)$
10:    unpad $output\_cmplx$
11:    $output \leftarrow concat(output\_cmplx.real, output\_cmplx.imag)$     ▷ complex to real value
12:    **return** $output$
13: **end procedure**

---

The linear map in Alg. 3 uses $setup\_weight$ per pixel to transform the input channels to output channels per pixel, which can be expressed as follows in Einstein summation convention:

$$OUT_{b,o}[x,y] = IN_{b,i}[x,y]W_{i,o}[x,y] \tag{16}$$

where $b$ denotes the batch index, $o$ is the output channel index, $i$ is the input channel index, and $x$, $y$ are the spatial indices. Therefore, we observe that $setup\_weight$ has a size of $C^2n^2$, which restricts the application of FNN to large-scale problems. Fortunately, it is only required for certain coarse levels in the multigrid hierarchy.

## E  A COMPUTATIONAL INEFFICIENCY ISSUE

A drawback of the FNN is the unexpected inefficiency in computing the linear map operation (see Eq. 16) in the frequency domain. The computational complexity of this operation is $\mathcal{O}(C^2n^2)$ for a $n \times n$ grid with $C$ channels, which should ideally be as swift as a convolution with $1 \times 1$ kernel. However, our timing tests reveal that this operation takes even longer than the FFT2d in PyTorch. This issue is less apparent in FNO-like methods, which consider significantly fewer modes. Consequently, future work should focus on developing a more efficient implementation of the linear map in FNN.

## F  SCALABILITY TEST DETAILS

We conduct scalability tests on 50 unseen data with varying grid sizes and angular frequencies. The results, depicted in Fig. 8, reveal that the iterations required for convergence vary across

different data, a characteristic typical of the Helmholtz equation. Indeed, certain speed (or slowness) distributions can exacerbate the ill-conditioned nature of the Helmholtz equation, leading to trapping phenomena Graham et al. (2019); Ralston (1971).

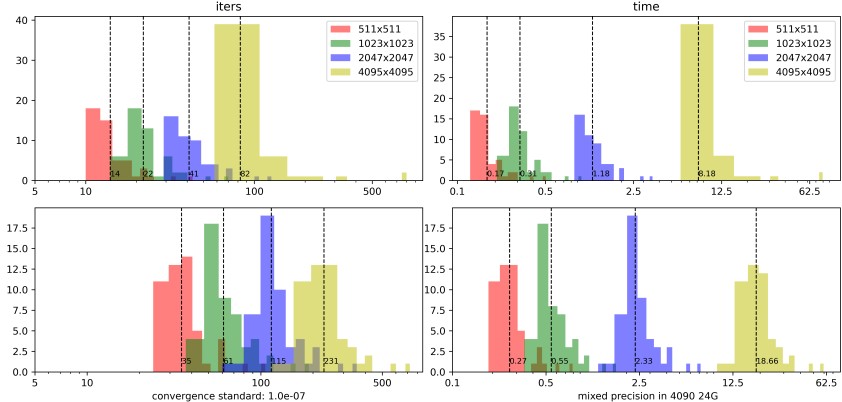

Figure 8: Scalability test over 50 unseen data. The upper row is the results of MGCFNN, and the lower row is the results of MGCNN.

## G    GENERALIZATION TO OTHER DISTRIBUTIONS

We further investigate the generalization of MGCFNN to other distributions. In the experiments, we apply the model trained on the CIFAR-10 dataset to the STL-10 dataset, which has a higher image resolution of $96 \times 96$, and to the benchmark Marmousi data, which has a resolution of $362 \times 1101$. The results are presented in Table 6, where MGCFNN demonstrates good generalization performance with a slight increase in number of solving iterations. This increase is expected, as a more complex medium results in more optic rays, which influence the convergence of the iterative solver Galkowski et al. (2024). Figures 9a and 9b illustrate the single source results on a STL-10 data and Marmousi model, respectively.

Table 6: Generalization results of number of iterations on STL-10 dataset and Marmousi. The model is trained on CIFAR-10 dataset.

| rtol=1E-7, iters | | standalone | | | GMRES | | |
|---|---|---|---|---|---|---|---|
| $\omega$ | size | CIFAR-10 | STL-10 | Marmousi | CIFAR-10 | STL-10 | Marmousi |
| $80\pi$ | $511 \times 511$ | 14 | 39 | 26 | 12 | 32 | 21 |
| $160\pi$ | $1023 \times 1023$ | 22 | 41 | 35 | 20 | 33 | 29 |
| $320\pi$ | $2047 \times 2047$ | 41 | 59 | 52 | 35 | 51 | 42 |
| $640\pi$ | $4095 \times 4095$ | 83 | 109 | 92 | 72 | 94 | 79 |

## H    MODEL ARCHITECTURE SETTINGS

Before discussing the architecture settings, it is important to note that the number of points per wavelength is a critical factor in determining the model architecture. For problems smaller than $511 \times 511$, there are about 6 points per wavelength, while for larger problems in scalability tests, there are approximately 12 points per wavelength. We set the number of multigrid levels such that the coarsest level has around 3 points per wavelength, and then add more levels to optimize the results. For MGCFNN, the level with around 3 points per wavelength is also the starting level where the FNN is used in the solve phase network. We find that using FNN in the two coarsest levels is beneficial. Additionally, to fully utilize GPU resources, we set a larger number of channels for smaller problems. Detailed model architecture settings for the solve phase are presented in Tables 7 and  8.

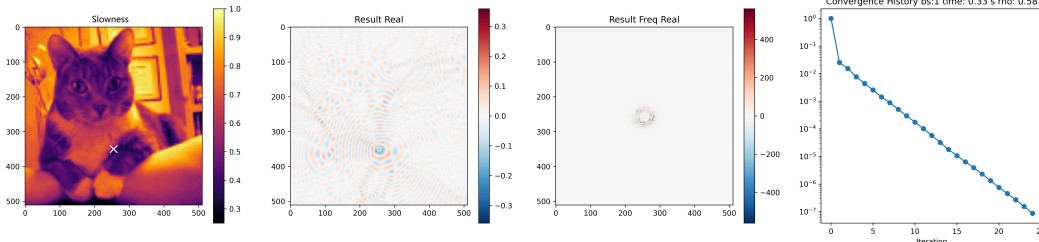

(a) Single source wave with $\omega = 80\pi$, in a slowness field generated from a STL-10 data.

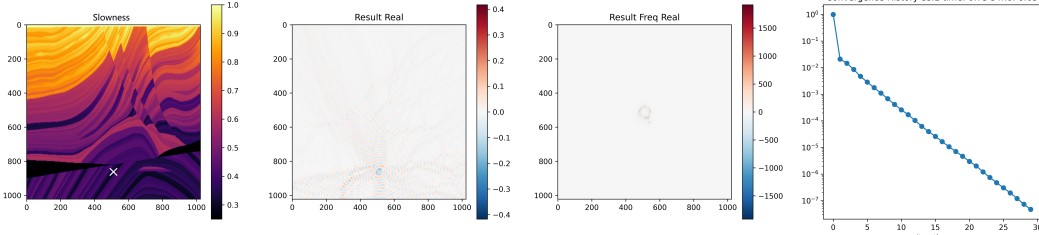

(b) Single source wave with $\omega = 160\pi$, in a slowness field from Marmousi.

Figure 9: From left to right: slowness field $\kappa(x)$, real value of wave field and wave field in frequency domain, iterative solving history. The cross symbol in the first column denotes the position of source.

Table 7: Solve phase settings for around 6 points in a wavelength.

| Model | Settings |
|---|---|
| MGCFNN, MGFNN | 32 channels, 1, 2, 4 sweeps for level 1, 2, 3 |
| FNN | 32 channels, 8 sweeps for one level |

Table 8: Solve phase settings for around 12 points in a wavelength.

| Model | Settings |
|---|---|
| MGCFNN | 12 channels, 1, 2, 4, 8 sweeps for level 1, 2, 3, 4 |
| MGCNN | 12 channels, 1, 2, 4, 8, 16 sweeps for level 1, 2, 3, 4, 5 |

For the kernel sizes in both the solve and setup phases, CNNs within ResNets use a kernel size of 5, while the restriction and prolongation CNNs use a kernel size of 3. All skip add operations are performed with a kernel size of 1. In the setup phase, we set two sweeps for each level, and the number of channels is consistent with the solve phase to ensure network compatibility.

## I   TRAINING SETTINGS

Table 9: Training settings.

| Parameter | Value | Description |
|---|---|---|
| epochs | 120 | – |
| num_data | 10000 | number of data used in training |
| lr | 0.001 | initial learning rate |
| optimizer | Adam | scheduler: step_size=6, gamma=0.8 |
| batch_size | 10 | – |

