# OpenReview forum: "MGCFNN: A Neural MultiGrid Solver with Novel Fourier Neural Network for High Wave Number Helmholtz Equations"
_ICLR.cc/2025/Conference — ICLR 2025 Poster_

### Official Review · Reviewer_q7vM · 2024-11-02

**Soundness:** 3
**Presentation:** 2
**Contribution:** 3
**Rating:** 6
**Confidence:** 4

**Summary:**

This paper introduces MGCFNN, an AI-based solver tailored for high wavenumber Helmholtz equations, which are  challenging, especially in heterogeneous media. Building on the MGCNN framework, the authors incorporate a Fourier neural network (FNN) to improve solution efficiency and accuracy, leveraging FNN's CNN-like structure for faster propagation of source influence. Experimental results reveal MGCFNN's promising performance in supervised learning compared to existing neural operator methods, as well as notable scalability improvements in unsupervised settings. An ablation study underscores the value of the multigrid hierarchy and FNN, with the solver achieving optimal convergence up to $k\approx 2000$.

**Strengths:**

1. The framework presented in this work, with its division into setup and solve phases, is both well-structured and innovative.
2. The paper introduces a novel Fourier neural network (FNN) that enhances the solver’s ability to manage high wavenumber problems efficiently, particularly within heterogeneous media, providing a significant advancement over traditional CNN-based approaches.
3. The manuscript includes multiple numerical examples that demonstrate improved performance over existing neural operator architectures. Additionally, the results in Tables 3 and 5 are particularly compelling and highlight the model's effectiveness.

**Weaknesses:**

1. The paper would benefit from a more detailed discussion of its novelty relative to prior work on MGCNN, explicitly highlighting the advancements MGCFNN brings over previous frameworks.
2. While the motivation appears intuitively reasonable, adding solid mathematical insights to support the design of MGCFNN would strengthen the rationale and deepen the reader’s understanding of the architectural choices.
3. The explanation in Section 3.1 lacks clarity, particularly regarding its connection to Sections 3.2 and 3.3. Providing clearer linkages between these sections would help readers follow the logic and flow of the methodology.

**Questions:**

In addition to the points noted in the "Weaknesses" section, I have the following questions:
1. Tables 1 and 2 present results with $\omega=80\pi$. How do the results compare at $\omega=640\pi$, given that the primary goal of this work is to address high-wavenumber cases?
2. The numerical results in Table 4 do not show a significant difference between MGFCNN and MGFNN. Additional arguments and comparisons between these two methods would help clarify their respective advantages.

---

> ### Author Response · Authors · 2024-11-18
> **MGCFNN vs. MGCNN**
>
> 1. In our paper, MGCNN has been modified to address the challenges posed by the high wavenumber Helmholtz equation. The original MGCNN is not well-suited for this problem, as we have tried in earlier tests, where it performs poorly. The modification, which includes key architectural changes—such as no weight sharing between levels, a fixed number of levels, and increased work on coarser levels—is also the aspect of our contribution.
>
> 2. The primary novelty of our approach lies in the FNN kernel, which addresses the limitations of the multigrid hierarchy for high wavenumber problems. FNN works to speed up high frequency wave propagation at coarser levels, where the coarse grid is nearly insufficient to capture the high frequency features. The integration of FNN in MGCFNN significantly reduces the number of iterations, cutting it by half compared to the modified MGCNN under the GMRES iterative framework, and providing an even greater advantage in a standalone iterative framework.
>
> We have revised **Section 3** to emphasize the complementary relationship between the multigrid hierarchy and the FNN kernel.
>
> Furthermore, The full potential of the FNN has yet to be explored. The computational inefficiency of the FNN’s linear transformation should not be the issue in future as its computational complexity is the same as a 1x1 kernel size convolution, and there should be better interpolation method in frequency domain of FNN. We have revised **Appendix E** and added **Remark A.1** (line 725) to clarify these points.

---

> > ### Comment · Reviewer_q7vM · 2024-11-19
> >
> > Thank you for the responses and results. The provided materials addressed most of my concerns, so I have increased the score.

---

> > > ### Author Response · Authors · 2024-11-19
> > >
> > > Thank you for your updated feedback and for taking the time to review our responses and additional results. We appreciate your constructive comments, which have helped us improve our work.
> > > If there are any further questions or concerns, we would be happy to address them.

---

> ### Author Response · Authors · 2024-11-18
> **Mathematical Insights**
>
> We deeply agree with the reviewer's suggestion to provide mathematical insights into the design of the FNN kernel.
> 1. In **Appendix A**, we provide a mathematical comparison between the damped wave equation and the FNN kernel, which offers insights into the design of the FNN for accelerating wave propagation.
>
> 2. The FNN kernel also bears connections to traditional spectral methods, such as the Born series method. We have stated these connections in **Appendix B** of the revised manuscript.

---

> ### Author Response · Authors · 2024-11-18
> **Section 3 Writing**
>
> We appreciate the reviewer's suggestion to improve the writing in **Section 3**. We have made the following changes to better link the subsections in **Section 3** in the revised manuscript:
> 1. In the revised manuscript, we emphasize FNN as the solve phase component to transport the source influence.
> 2. Additionally, we emphasize both FNN and multigrid hierarchy have their own limitations and it is their complementary combination that result in the success of MGCFNN.

---

> ### Author Response · Authors · 2024-11-18
> **Comparison frequency with Neural Operators**
>
> We present results with $\omega=80\pi$ is for the purpose of clear and practical comparison.
>    1. A single application of the neural network yields low accuracy for $\omega = 80\pi$, as shown in Table 1 and Table 2. For higher frequencies, the accuracy drops even further, which may not provide meaningful insights. This is also why we first present the results for $\kappa_{\min} = 0.75$ in Table 1 to reduce the problem's difficulty and make the comparison clearer.
>    2. Additionally, the FNO methods in comparison does not scale with higher frequency, and high frequency needs higher resolution grid to discretize. Training directly on $\omega = 640\pi$ is then not feasible due to the high computational cost, memory consumption, and the difficulty in generating the required supervised dataset.

---

> ### Author Response · Authors · 2024-11-18
> **MGCFNN vs. MGFNN**
>
> We acknowledge that the difference between MGCFNN and MGFNN in the ablation study (Table 4) is not significant. Firstly pure FNN-based methods, such as MGFNN and FNN, are less suitable for large problems due to the memory consumption of their setup weights. Secondly, the solve time difference becomes more pronounced for larger problems. Below, we present scalability test results up to a 1024x1024 mesh to highlight the differences between MGFNN and MGCFNN. However, we have not included these results in the revised manuscript, as we used different model configurations (see **Appendix H**) for the small and scalability tests, and we wish to avoid confusion regarding the model configurations.
>
> | Size       | MGFNN Iterations | MGFNN Time | MGCFNN Iterations | MGCFNN Time |
> |------------|-------------------|-------------|------------------|------------|
> | 256x256    | 13                | 0.116       | 15               | 0.111      |
> | 512x512    | 21                | 0.268       | 23               | 0.206      |
> | 1024x1024  | 39                | 1.747       | 43               | 1.096      |
>
> In the revised manuscript, we have emphasized the point in ablation study and added **Remark 3.3** (line 312) to discuss the architecture settings which is related to this question.

---

### Official Review · Reviewer_yYix · 2024-11-04

**Soundness:** 3
**Presentation:** 3
**Contribution:** 3
**Rating:** 6
**Confidence:** 4

**Summary:**

This work presents an AI-driven PDE solver that specifically utilizes a Fourier neural network. By independently processing the coefficient and source term using a modified spectral convolution operation, it effectively handles high wavenumber Helmholtz equations.

**Strengths:**

* The modification of spectral convolution successfully handles full-mode information
* MGCFNN effectively handles the linearity of source terms
* It demonstrates excellent scalability with high-wavenumber problems

**Weaknesses:**

* Does MGCFNN work well only for high-frequency problems?
* Can MGCFNN be applied to other PDEs in operator learning?

**Questions:**

* In Figure 2, the visualization of the network architecture is difficult to understand

---

> ### Author Response · Authors · 2024-11-18
> **Helmholtz vs Other PDEs**
>
> 1. It is an insightful question, as the application of MGCFNN is closely related to the philosophy of multigrid hierarchy and FNN.
>    1. **Philosophy of Multigrid Hierarchy and FNN**: The multigrid hierarchy is effective for handling global features, especially when coarser levels can capture these features. Traditional multigrid methods are typically the best choice for several PDEs, including low-frequency Helmholtz equations. However, they are less effective for high-wavenumber Helmholtz equations. This limitation motivated us to develop the FNN kernel, which accelerates wave propagation at coarser levels for high-wavenumber problems. Therefore, the multigrid hierarchy and FNN are complementary in MGCFNN, and the choice of specific architecture depends on the wave number of the problem. We have revised **Section 3** to emphasize this point.
>    2. **Low frequency Cases and Other PDEs** MGCFNN employs a multigrid hierarchy and should be applicable to other PDEs or low-frequency cases.
>       1. We have tested MGCFNN with one more level on the same slowness field as in **Fig. 4b** with $\omega=16\pi$. The iteration number was reduced from 19 to 7, and the computation time decreased from 0.15 seconds to 0.09 seconds.
>       2. We have also tested the MGCFNN model on a convection-diffusion equation with varying diffusion coefficients over a 256x256 mesh and mesh Reynolds number in the range [1, 1000]. The model takes only 0.013 seconds and 3 iterations to solve.
>    However, using FNN is not necessary in these cases, as coarser levels effectively capture global features. This is consistent to the **Ablation Study**, where MGFNN, using FNN in all levels, slightly reduces iterations compared to MGCFNN while consuming more memory and time. We have added this discussion in **Remark 3.3** (line 312) of the revised manuscript to clarify this point.
> 2. **Focus on Helmholtz Equation**: Although various types of equations exist, the high wavenumber Helmholtz equation poses significant challenges for traditional iterative solvers. Furthermore, to the best of our knowledge, no pure neural network methods have effectively solved this equation under extreme high wavenumber and non-absorbent medium conditions. Therefore, we focus on this problem to validate the efficacy of pure neural network approaches for extremely difficult problems.
>
> In summary, the MGCFNN architecture and the use of FNN are problem-dependent, which is common in iterative methods. Our goal in this paper is not to establish a new SOTA for all PDEs, but rather to address the specific challenges posed by high-wavenumber Helmholtz equations. As such, we compare our method with recent specialized pure AI methods and AI-enhanced traditional methods for high-wavenumber Helmholtz equations.

---

> ### Author Response · Authors · 2024-11-18
> **Explanation of the FNN Computation Process**
>
> To shorten the main content length, we combined two cases of the FNN weights setup process into a single plot, which may have caused some confusion. In the revised manuscript, we have itemized the description of FNN in **Section 3.1** to make it easier to understand Figure 2 with the accompanying textual explanation. Additionally, we have included pseudocode in **Appendix D** to clarify the process further. We hope these changes will help you better understand the network architecture. If you have any further questions, please let us know.

---

> ### Comment · Reviewer_yYix · 2024-11-25
>
> After reading both the review and the revised manuscript, I believe my concerns have been adequately addressed. I sincerely appreciate the authors' efforts.

---

### Official Review · Reviewer_pjJn · 2024-11-08

**Soundness:** 3
**Presentation:** 3
**Contribution:** 2
**Rating:** 6
**Confidence:** 2

**Summary:**

The paper introduces MGCFNN, a multigrid-hierarchical neural solver designed to address high-wavenumber Helmholtz equations, which are challenging for both traditional and AI solvers. The authors propose a Fourier Neural Network (FNN), an enhancement of spectral convolution operations, specifically to manage full-mode information and accelerate wave propagation during solving. The authors evaluate MGCFNN on both supervised and unsupervised tasks, comparing its performance against various neural operator learning methods.

**Strengths:**

1. The paper is well written and easy to follow
2. The paper presents extensive experimental validation, including scalability tests and comparisons with recent operator learning methods.
3. This is interesting idea for generalizing FNO to a more generic architecture.

**Weaknesses:**

1. Comparison is limited. The paper mainly focuses on a few high-wavenumber scenarios for Helmholtz equations. There are a broad range of equations worth to be evaluated.
2. Hyperparameters: the paper lacks discussion on the choice and tuning cost of their hyperparameters, such as number of multigrid levels, number of Fourier modes, etc.

**Questions:**

1. Can you provide more comparison experiments especially with other types of equations?

2. Can you provide details on your hyperparameter selection and cost of tuning and selection?

---

> ### Author Response · Authors · 2024-11-18
> **Helmholtz vs Other PDEs**
>
> 1. It is an insightful question, as the application of MGCFNN is closely related to the philosophy of multigrid hierarchy and FNN.
>    1. **Philosophy of Multigrid Hierarchy and FNN**: The multigrid hierarchy is effective for handling global features, especially when coarser levels can capture these features. Traditional multigrid methods are typically the best choice for several PDEs, including low-frequency Helmholtz equations. However, they are less effective for high-wavenumber Helmholtz equations. This limitation motivated us to develop the FNN kernel, which accelerates wave propagation at coarser levels for high-wavenumber problems. Therefore, the multigrid hierarchy and FNN are complementary in MGCFNN, and the choice of specific architecture depends on the wave number of the problem. We have revised **Section 3** to emphasize this point.
>    2. **Low frequency Cases and Other PDEs** MGCFNN employs a multigrid hierarchy and should be applicable to other PDEs or low-frequency cases.
>       1. We have tested MGCFNN with one more level on the same slowness field as in **Fig. 4b** with $\omega=16\pi$. The iteration number was reduced from 19 to 7, and the computation time decreased from 0.15 seconds to 0.09 seconds.
>       2. We have also tested the MGCFNN model on a convection-diffusion equation with varying diffusion coefficients over a 256x256 mesh and mesh Reynolds number in the range [1, 1000]. The model takes only 0.013 seconds and 3 iterations to solve.
>    However, using FNN is not necessary in these cases, as coarser levels effectively capture global features. This is consistent to the **Ablation Study**, where MGFNN, using FNN in all levels, slightly reduces iterations compared to MGCFNN while consuming more memory and time. We have added this discussion in **Remark 3.3** (line 312) of the revised manuscript to clarify this point.
> 2. **Focus on Helmholtz Equation**: Although various types of equations exist, the high wavenumber Helmholtz equation poses significant challenges for traditional iterative solvers. Furthermore, to the best of our knowledge, no pure neural network methods have effectively solved this equation under extreme high wavenumber and non-absorbent medium conditions. Therefore, we focus on this problem to validate the efficacy of pure neural network approaches for extremely difficult problems.
>
> In summary, the MGCFNN architecture and the use of FNN are problem-dependent, which is common in iterative methods. Our goal in this paper is not to establish a new SOTA for all PDEs, but rather to address the specific challenges posed by high-wavenumber Helmholtz equations. As such, we compare our method with recent specialized pure AI methods and AI-enhanced traditional methods for high-wavenumber Helmholtz equations.

---

> ### Author Response · Authors · 2024-11-18
> **Hyperparameters Tuning**
>
> 1. **Fourier Modes and Multigrid Levels**: There is no need to consider the number of Fourier modes, as our strategies allow to use all modes in the Fourier space. The number of multigrid levels is determined by the problem's wave number and grid size. Specifically, we ensure that coarser levels are used until there are 3 points per wavelength. After that, we incrementally add more levels to assess whether they provide additional benefits. We have added **Remark 3.3** (line 312) and revised the **Appendix H** (original E) in the revised manuscript to clarify this point.
>
> 2. **Hyperparameter Tuning on Small Grids**: The network is consistent across different scales, which allows us to perform hyperparameter tuning on smaller grids. In Table 1, we list the training time per epoch, which is about one minute—faster than any other models in the table. The training process is of strong offline feature, as the trained network generalizes well across different scales, sources, and wave numbers. Consequently, the cost of hyperparameter tuning may not be a significant issue.

---

### Official Review · Reviewer_T4nv · 2024-11-08

**Soundness:** 2
**Presentation:** 2
**Contribution:** 2
**Rating:** 6
**Confidence:** 4

**Summary:**

Authors develop a hybrid iterative solver for Helmholtz equation. A proposed technique is a combination of multigrid-like architecture with a specially modified Fourier Neural Operator and a classical modified Richardson iteration. As I understand (although the description is not complete as I discuss below), authors apply a neural network to produce parameters of linear operators from PDE data and later apply these linear operators within a multigrid-like architecture. The resulting solver shows promising results on several high frequency problems.

**Strengths:**

1. The problem authors consider is well motivated, since it is known that classical iterative methods suitable for elliptic problems perform poorly on the Helmholtz equation.
2. A proposed method is a combination of multigrid and machine learning techniques, so, potentially, it scales well, is consistent and may be more robust than purely ML-based solvers.
3. Benchmarks authors propose cover especially challenging problems with high wavenumbers.

**Weaknesses:**

An article has several weaknesses which I briefly describe below:
1. The description of the method is not entirely clear.
2. Not all relevant baselines are covered.
3. The dataset is chosen in a very specific manner, making generalization hard to access.
I elaborate on these and other issues in the next section.

**Questions:**

1. The description of the method is not entirely clear.
   Authors provide descriptions of their approach in various places in the article. In my opinion these descriptions are confusing and incomplete. It is possible that other readers will find descriptions insufficient as well. Below is a list of questions on architectures authors used:
   1. Appendix E contains parameters of MGCFNN and MGFNN. Are these parameters for the setup phase network or for the solve phase network (or both)?
   2. Does the solve phase network have other parameters than the one produced by the setup phase network? If the answer is yes, what are those parameters?
   3. Is it the case that authors explicitly used residual only on the fine level, i.e., in equation (9) network implement linear operator B?
   4. Authors write about inefficiency of linear transformation in appendix D. From the number of operations authors provide it seems that this is a convolution with kernel size 1. Is that the case? Can the authors provide a number of floating point operations their architectures perform? This is highly relevant for comparison with classical iterative methods.
1. Not all relevant baselines are covered.
   1. Many well developed classical methods are available for the Helmholtz equation. It looks problematic that authors do not compare with any of them. For some reason all baselines are other ML approaches. I suggest the authors provide results for at least shifted-Laplace preconditioners and ideally with the optimized Schwartz method https://arxiv.org/abs/1610.02270. It is especially interesting to consider time needed to achieve desired accuracy since ML-based preconditioners are typically more computationally involved.
   2. Since authors consider only 2D cases, sparse direct solvers are also relevant baseline (see section 4 in https://arxiv.org/abs/1610.02270). I suggest providing solution time for modern CUDA implementation of sparse LU, preferable using freely available SuiteSparse solver https://people.engr.tamu.edu/davis/suitesparse.html.
   3. Sparse algebraic ML-based preconditioners (often GNN-based) form another class of relevant baselines, for example, https://arxiv.org/abs/2405.15557, https://arxiv.org/abs/2305.16432, etc
   4. Completely nonlinear methods based on flexible Krylov methods are also a valid solution approach https://arxiv.org/abs/2401.02016v2, https://arxiv.org/abs/2402.05598.
2. Dataset construction and generalization.
   First, the dataset authors used is detached from practically relevant applications of the Helmholtz equation. Second, a main dataset is formed from CIFAR-10, which has only 32x32 resolution which is interpolated to access performance on grids with higher resolutions. The troubling part is that fields, produced in such a way, have fixed frequency content, which is already known at the lowest resolution. Given that, it does not allow one to test whether the preprocessing network used to construct linear operators can generalize well to the problems of other resolutions. I suggest authors provide generalization tests where $k(x)$ contains higher frequencies on fine grids.
4. The fixed number of levels is not completely satisfactory.
   Authors used multigrid with a fixed number of layers. This means, with the increase in resolution the performance of the solver will degrade. I can suggest authors to empirically demonstrate that this is not the case. A possible experiment is to train architecture with two levels (i.e., the smallest possible) on a small grid and after that apply it to grids with increasing resolutions (much larger than the original grid). If one observes no degradation, the fixed number of levels is not a problem.
5. Is a comparison with NO fair?
   Neural operators have only access to input-output pairs, while the method in the current article explicitly uses discretisation. I think it is appropriate to explicitly discuss this distinction.

---

> ### Author Response · Authors · 2024-11-18
> **Model Architecture Related Questions**
>
> Due to page constraints, we moved some of details to the appendix in order to focus only on the core aspects in the main text. Below are our responses to the reviewer’s questions and we clarify these points further in the revised manuscript.
>
> 1. **Setup Phase Hyper-parameters**: We acknowledge that the description of the setup phase hyper-parameters was not clear. Actually, the kernel size and number of channels of setup components are the same as corresponding solve phases' components.
>
> 2. **Solve Phase Hyper-parameters**: The tables in **Appendix H** (originally Appendix E) is for hyper-parameter settings in the solve phase.
>
> 3. **Residual Computation**: The network works as solver or preconditioner operator $B$ in an iterative framework, and there is no residual computation within the network itself. The residual computation is necessary when using the iterative framework or doing unsupervised learning.
>
> 4. **Linear Transformation and Computational Complexity**: The computational complexity of a linear transformation is similar to that of a convolution with a kernel size = 1. The transformation, however, is performed using varying kernel weights across the domain. In Einstein summation convention, the operation can be expressed as:
>    $$ OUT_{b,o}[x,y] = IN_{b,i}[x,y] W_{i,o}[x,y] $$
>
>    where $b$ denotes the batch index, $o$ is the output channel index, $i$ is the input channel index, and $x$, $y$ are the spatial indices.
>    - We have tried the Pytorch functions *fvcore.nn.flop_count.FlopCountAnalysis* and *torchprofile.profile_macs* to count the flops, but it seems that *fft* operation is not supported. While the number of floating point operations is an important metric, we focus primarily on absolute time cost, as matrix-vector multiplication on GPUs is highly optimized.

---

> ### Author Response · Authors · 2024-11-18
> **Comparison Related Questions**
>
> We appreciate the reviewer's suggestions regarding the comparison methods. Below are our responses to the reviewer's questions, and we have added an additional **Comparison Methods** (starting from line 326) part and **Remark 4.1** (line 504) in the **Numerical Experiments** section to clarify some points in the revised manuscript.
>
> 1. **Traditional Iterative Methods**: We agree that there are specially developed classical methods for solving the Helmholtz equation, and the shifted-Laplacian method has been used as a baseline in our referenced studies. However, as shown in these studies, the shifted-Laplacian method performs significantly worse. Given this, we did not include shifted-Laplacian but only their AI or AI-enhance methods in our **Scalability Comparison** subsection. Actually, one of our comparison method, Wave-ADR-NS solver, is an AI-enhanced traditional method which build partly on the wave-ray method, which is another traditional method aimed at Helmholtz equations. We are also aware of other advanced parallel traditional solvers (e.g., source transfer(https://epubs.siam.org/doi/10.1137/18M1196170), trace transfer(https://www.sciencedirect.com/science/article/pii/S0021999122000420)). These solvers are over 5x slower than our method under similar wave number and grid size conditions.
>
> 2. **Sparse LU Methods**:  We understand that sparse LU is a good method for difficult 2d problems. We find a GPU-accelerated sparse LU solver PanguLU(https://dl.acm.org/doi/pdf/10.1145/3581784.3607050), which outperforms the well-known SuperLU solver. In their paper, PanguLU takes 2.37 seconds for the *ecolopy1* matrix (5-point stencil for a $1000\times1000$ mesh), while MGCFNN takes only around 0.31 seconds for a $1023\times1023$ mesh.
>
> 3. **GNN-Based Methods**: We are aware of some GNN-based methods for solving PDEs. Actually, GNNs are more suitable for unstructured grid problems. These methods do not leverage the structured grids and are less computationally efficient than CNN-based methods. Therefore, we do not consider it is fair to compare them with our methods.
>
> 4. **Krylov Methods and Nonlinear Networks**: We acknowledge that completely nonlinear methods as nonlinear preconditioner in flexible Krylov methods, as done in our comparison methods, are enable to improve accuracy with more iterations. However, it involves more effort on dataset to improve the nonlinear network's generalization for different RHS distributions. To provide a direct comparison with most fully nonlinear methods, we performed a one-step solution comparison with them to illustrate the learning efficiency of our methods. Our focus is on comparison with specialized methods for the Helmholtz equation under extreme high wave number conditions.

---

> ### Author Response · Authors · 2024-11-18
> **Dataset Related Questions**
>
> 1. **Ultrasonic CT Dataset**: There are results on a 480x480 dataset derived from an ultrasonic CT competition in **Section 4.4**, which is closely related to practical applications. On the other hand, CIFAR-10 contains more classes of items, which is the same dataset used in the comparison methods' papers.
>
> 2. **Generalization and Dataset Frequency**: We appreciate the concern regarding generalization and dataset frequency. We conducted generalization tests on other datasets, including the STL-10 dataset (96x96 size) and the benchmark Marmousi dataset (362x1101 size). We have added an additional **Appendix G** to show the results on these datasets in the revised manuscript. There is some reasonable increase in the number of iterations.

---

> ### Author Response · Authors · 2024-11-18
> **Fixed Number of Levels Related Questions**
>
> We understand the reviewer's concern regarding the fixed number of levels in the multigrid hierarchy. Fixed levels are used in the context of higher wavenumber problems, which result in larger problem sizes requiring suitable discretization. The use of a fixed level strategy, such as the three-grid cycle~\cite{e92a9f64927e8ee73f5b10b256c7c3bef8016550}, is common in multigrid methods for high wavenumber Helmholtz problems. This is because coarser levels are not effective in approximating higher-frequency errors, which motivates our design of the FNN to accelerate wave propagation at some coarse levels. We have added **Remark 3.3** (line 312) in the revised manuscript to clarify the choice of the number of levels.

---

> ### Author Response · Authors · 2024-11-18
> **Neural Operator Comparison Related Questions**
>
> 1. We compare Neural Operator methods under the same supervised setting, where only access to input-output pairs is offered. The discretization is only used to generate the supervised dataset.
> 2. Basically, the discretization is needed when residual calculation is required, which is necessary for iterative framework or unsupervised learning loss.

---

> ### Comment · Reviewer_T4nv · 2024-11-21
> **Sparse LU Methods**
>
> ```
> We find a GPU-accelerated sparse LU solver PanguLU(https://dl.acm.org/doi/pdf/10.1145/3581784.3607050), which outperforms the well-known SuperLU solver. In their paper, PanguLU takes 2.37 seconds for the ecolopy1 matrix (5-point stencil for a $1000\times1000$ mesh), while MGCFNN takes only around 0.31 seconds for a $1023\times1023$ mesh.
> ```
>
> I was able to read the article and locate the result authors mentioned. However, the authors of PanguLU report this result for A100 and probably for double precision (unclear from the article), whereas 0.31 seconds for MGCFNN is reported with single precision on RTX 4090.
>
> 1. Can the authors run sparse linear solvers (of choice, preferable https://developer.nvidia.com/cholmod) on their hardware with the same precision and report results?
> 2. Is the setup phase time included in these 0.31 seconds or is it only a solve time?

---

> > ### Author Response · Authors · 2024-11-22
> >
> > 1. We reported results on the A100 in the **Scalability Comparison** section 4.3 in Table 3. The results are close to those of the RTX 4090, and for a 1023x1023 mesh, the A100 is marginally faster at 0.30 seconds.
> > 2. PangLU uses double precision. We tested our MGCFNN with double precision, and the time for a 1023x1023 mesh is 1.08 seconds.
> > 3. The setup phase time is included in the 0.31 seconds reported. It is marginal compared to the solve phase time, as there are more iterations in the solve phase.
> > 4. We have attempted to install CHOLMOD but encountered some issues. We are working on it and will report the results as soon as possible.

---

> ### Author Response · Authors · 2024-11-23
> **Comparison with CHOLMOD (SuiteSparse)**
>
> We have successfully installed CHOLMOD (version 5.3.0) in SuiteSparse (version 7.8.3). It runs faster than PanguLU's reported result on the ecology1 matrix, taking around 1.95 seconds. However, it takes much more time on our problems, as our systems are complex-valued and it factorizes $A*A' + \beta*I$. We attempted to use a GPU to accelerate the computation, but encountered several issues, which are listed below.
>
> 1. We noticed low GPU utilization. In their paper(https://doi.org/10.1016/j.parco.2016.06.004), GPU acceleration is beneficial for cases where dense operations dominate and there are many fill-ins in the LU factorization, which is not the case for our problem system.
> 2. CHOLMOD consumes lots of memory on the GPU and does not work for large-scale problems.
> 3. CHOLMOD only allows us to use double precision, which limits our comparison on same computation precision.
> 4. The improvement on the GPU is not as significant as running on the CPU with single precision.
> 5. Our problem system is very ill-conditioned. Running on the GPU results in NaN values for the 2047x2047 scale and around 1E-4 relative residual for the 1023x1023 scale. Running on the CPU with double precision yields around 1E-7 relative residual, while single precision on the CPU results in around 1E-6 relative residual.
>
> We ultimately tested CHOLMOD on a high-performance workstation with an Intel(R) Xeon(R) Gold 6458Q CPU and tuned the number of threads to achieve the best performance for each scale. The results are shown in the following table:
>
> | omega  | grid       | MGCFNN time(s) | CHOLMOD time(s) | speedup |
> |--------|------------|----------------|------------|---------|
> | 80pi   | 511x511    | 0.16           | 0.49       | 3.1     |
> | 160pi  | 1023x1023  | 0.30           | 8.88       | 29.5    |
> | 320pi  | 2047x2047  | 1.15           | 38.91      | 33.8    |
> | 640pi  | 4095x4095  | 8.55           | 183.61     | 21.5    |
>
> Our MGCFNN is still faster than CHOLMOD on the CPU with single precision. We have updated the paper with these results in **Scalability Comparison** section 4.3.

---

### Author Response · Authors · 2024-11-26
**Gratitude for the Reviewers' Feedback and Support**

We would like to express our heartfelt gratitude to all reviewers for their time, effort, and constructive feedback. Your valuable insights and suggestions have greatly contributed to enhancing the quality of our work. Thank you once again for your thoughtful and encouraging comments throughout the review process.

---

### Meta-Review · Area_Chair_GTtc · 2024-12-22

**Metareview:**

The paper proposes a learnable multigrid solver, tailored for high wavenumber Helmholtz equations, which always has been a challenge in numerical computations, and the solvers exhibit O(k) convergence. The problem considered is a 5-point discretization on a 2D grid: simple, but having practical applications. The description of the actual architecture and the loss function is quite vague: need to be improved in the final version, but I think the overall contribution is quite reasonable.

**Additional Comments On Reviewer Discussion:**

There has been a nice discussion between the authors and Reviewer T4nv which turned out to be very positive: additional comparisons and references added which confirm the effectiveness of the method.

---

### Decision · Program_Chairs · 2025-01-22

Accept (Poster)